# Engineering of Bio-Adhesive Ligand Containing Recombinant RGD and PHSRN Fibronectin Cell-Binding Domains in Fusion with a Colored Multi Affinity Tag: Simple Approach for Fragment Study from Expression to Adsorption

**DOI:** 10.3390/ijms22147362

**Published:** 2021-07-08

**Authors:** Amina Ben Abla, Guilhem Boeuf, Ahmed Elmarjou, Cyrine Dridi, Florence Poirier, Sylvie Changotade, Didier Lutomski, Abdellatif Elm’selmi

**Affiliations:** 1EBInnov^®^, Ecole de Biologie Industrielle, 49 Avenue des Genottes, 95000 Cergy, France; a.benabla@hubebi.com (A.B.A.); g.boeuf@hubebi.com (G.B.); c.dridi@hubebi.com (C.D.); 2Unité de Recherche Biomatériaux Innovants et Interfaces URB2i, Université Paris Sorbonne Nord, 74 Rue Marcel Cachin, 93017 Bobigny, France; florence.poirier@univ-paris13.fr (F.P.); changotade@univ-paris13.fr (S.C.); lutomski@univ-paris13.fr (D.L.); 3Plateforme de Production D’Anticorps et de Protéines Recombinantes, Institut Curie/CNRS UMR144, 75248 Paris, France; elmarjou@curie.fr

**Keywords:** bio-adhesive ligand, fusion protein, monitoring process, protein adsorption monitoring, cell adhesion

## Abstract

Engineering of biomimetic motives have emerged as promising approaches to improving cells’ binding properties of biomaterials for tissue engineering and regenerative medicine. In this study, a bio-adhesive ligand including cell-binding domains of human fibronectin (FN) was engineered using recombinant protein technology, a major extracellular matrix (ECM) protein that interacts with a variety of integrins cell-surface’s receptors and other ECM proteins through specific binding domains. 9th and 10th fibronectin type III repeat containing Arginine-Glycine-Aspartic acid (RGD) and Pro-His-Ser-Arg-Asn (PHSRN) synergic site (FNIII9-10) were expressed in fusion with a Colored Multi Affinity Tag (CMAT) to develop a simplified production and characterization process. A recombinant fragment was produced in the bacterial system using *E. coli* with high yield purified protein by double affinity chromatography. Bio-adhesive surfaces were developed by passive coating of produced fragment onto non adhesive surfaces model. The recombinant fusion protein (CMAT-FNIII9/10) demonstrated an accurate monitoring capability during expression purification and adsorption assay. Finally, biological activity of recombinant FNIII9/10 was validated by cellular adhesion assay. Binding to α5β1 integrins were successfully validated using a produced fragment as a ligand. These results are robust supports to the rational development of bioactivation strategies for biomedical and biotechnological applications.

## 1. Introduction

Engineering biomimetic motives delivering specific signals that direct cell function especially integrin-mediate cellular adhesion have emerged as promising approaches to improving cells-binding properties of biomaterials for tissue engineering and regenerative medicine [1]. Biomaterials are commonly used in different therapeutic strategies for tissue engineering and bone reparation [2,3]. Nevertheless, several limitations are still facing their biological integration. One major disadvantage is the lack of interaction with the surrounding environment leading to immunological complications [4]. These limitations have driven the need for the development of innovative approaches to promote rapid biomaterials and efficient bio-integration. One promising strategy is the immobilization of the extracellular matrix (ECM) proteins onto biomaterials’ surface to generate bio-activated surface miming the ECM functions [5,6]. Fibronectin (FN) is a key element of ECM through its contribution to structural integrity and biological function of tissues and organs [7]. This dimeric glycoprotein is composed of two identical polypeptides structurally stabilized by disulfide bonds at their C-termini [8]. It is found in a soluble form in liver and blood [9] and secreted by different cells such as endothelial cells and fibroblasts in insoluble form [10]. Several studies have underlined the considerable implication of FN in a variety of biological processes especially in cytoskeletal assembly, cell attachment, and migration [11,12], since its different domains are specific ligands for a variety of ECM proteins principally collagen [13,14] and heparin [15] as well as integrins cell surface receptors [16,17,18]. Therefore, controlling FN-integrins-specific interactions offers an innovative strategy to engineering biofunctionalized surfaces promoting cellular adhesion and stimulating cellular proliferation and migration [11,19,20]. So far, FN interactions with cell surface integrin’s receptors have been widely reported and later studies have shown the implication of a short peptide sequence Arg-Gly-Asp (RGD) localized in the cell-binding domain (FNIII10) as the major functional peptide responsible for the cell’s binding [21]. Although FN-RGD sequence (FNIII 10) binds to a wide range of integrins such α5β1 [22] and αvβ3 [23], additional synergic domains are required for promoting specific integrin interactions to FN [24]. Notably, it was established that the presence of an additional sequence Pro-His-Ser-Arg-Asn (PHSRN) localized in the 9th domain type III repeat (FNIII9) is required to act synergistically with the 10th domain in binding α5β1 integrin [25], which controls cellular attachment and migration, exchanges signals that prevent apoptosis and maintains cell-cycle progression [26].

Because of the expensive cost and low availability of native ECM proteins including FN [5], their use remains critical and limited. Other observations indicate the loss of conformation while proceeding to adsorption of native FN on several supports leading to a reduced cell receptor response [27]. To circumvent these drawbacks, researchers tended to engineer small recombinant protein fragments deriving from functional ECM protein domains ensuring oriented bioactivity with reduced cost and improved affinity [18,28] as well as enabling the design of novel combined recombinants fragments for additional activities [22,29,30,31]. In this case, the fusion tag technology has been extensively used for recombinant fragments production to allow efficient purification with reduced time and cost when compared to native ECM protein purification [32]. Affinity purification tags, such as a polyhistidine tag and streptavidin/biotin derivative systems, have become indispensable fusion partners for easy purification or immobilization of proteins [33,34]. Thus, fibronectin domains have been widely expressed with fusion partners for easy step purification using the Histidine (His)-tag [35] and Glutathion S-transferase (GST)-tag [31] or for enhancing bifunctionality using cadherin [36] and keratinocyte growth factors [37]. A challenging area in this field is the design of improved production strategy for high purity and robust characterization of a biomimetic recombinant fragment [38,39]. Recently, new fusion strategies have emerged for monitoring the production and characterization process and seems very promising. However, the marker whose tags are relatively less described in the literature and most current processes for recombinant protein production have limited monitoring capabilities. We previously reported the design of a Colored Multi Affinity Tag (CMAT) which contains a colored marker cytochrome b5 and two affinity purification tags: 10His (polyhistidine) and SBP (streptavidin binding protein). The key advantage of this technology is that enabling an easy monitoring of production and purification process and our previous results showed a high purity purification using the double-purification system [40].

The aim of this work is to produce the functional sites of the FN-cell-binding domains using recombinant protein technology as well as to develop a simple and efficient approach for its characterization and biological activity evaluation. Recombinant 9 and 10 type III fibronectin domains (rFNIII9/10) was expressed in *E. coli* in fusion with CMAT-tag reported in our previous studies [40,41]. This system has demonstrated its performance in real time monitoring of rFNIII9/10 production and purification steps allowing high degree of purity and improved evaluating strategies. The adsorption of rFNIII9/10 fragment onto microplate wells surfaces was evaluated using a novel method based on the CMAT tag. We further investigated the rFNIII9/10 biological activity by studying its capacity as a bio adhesive ligand and its interaction with integrins. α5β1 integrins were showed to successfully bind rFNIII9/10. Finally, engineered rFNIII9/10 was shown to be efficient in promoting gingival fibroblasts cell adhesion.

## 2. Results

In the present study, a functional adhesive ligand containing recombinant FN 9th and 10th type III repeats (rFNIII9/10) including RGD sequence and PHSRN synergic site was designed using standard gene cloning and plasmid construction techniques.

To quantitively and qualitatively monitor the rFNIII9/10 expression and to simplify characterization steps, the CMAT-tag [40,41] was used as the fusion partner.

Absence of mutation or alteration in the sequence used for CMAT-FNIII9/10 expression was confirmed by PCR, enzyme digestion, and DNA sequencing. Expression of the fusion protein was carried out using standard recombinant protein technology.

### 2.1. Production of Recombinant CMAT-FNIII9/10

*E. coli BL21* (DE3) pLysS was transformed with the recombinant pET15b holding the 9th and 10th type III repeats. During production, the soluble expression of rFHIII9/10 and biomass were evaluated in real time due to the CMAT fusion partner and was analyzed by SDS–PAGE. As we previously reported [40], the appearance of a red color was directly observed during the expression and the extraction steps of rFNIII9/10-tagged protein. Thereby, this red color property of cytochrome b5 provides a simple and powerful method for an accelerated screening of the optimal expression condition. Furthermore, early information about rFNIII9/10 solubility is accurately identifiable by following the coloration of the soluble fraction and the harvested cell pellet without resorting to the time-consuming steps of gel analysis (Figure A1). As we have demonstrated previously [40], a direct correlation was found between the red coloration intensity of fractions and produced protein with an optical density (OD) profile at 400 nm measurement of each sample. This system allows a rapid and efficient method to monitor protein expression at different points of time during production (Figure 1).

Recombinant fragment’s soluble expression increased with biomass increasing after induction (Figure 1A). Finally, rFNIII9/10 was produced with a yield of 200 µg per 100 mL LB culture.

### 2.2. Protein Purification and Detection

To purify the rFNIII9/10 produced fragment, double affinity chromatography purification using 10His tag and SBP tag was performed in this study through a CMAT fusion partner. Subsequent Coomassie blue-stained SDS-PAGE gel analysis revealed the presence of a protein band around 45 kDa in the purified fractions, consistent with the expected molecular weight of the fusion protein (CMAT-rFN). This peptide presents a reduced weight (45 kDa) compared to full FN (450 kDa) [9]. Short FN domains have been demonstrated to be less antigenic than longer proteins from which they derive [42] in addition to providing specific interaction with receptors to predict a specific cellular response [30]. Results presented in Figure 2 show residual contaminants when purification was performed using a 10His tag only. This was also consistent with Keefe et al. who showed a lower purification degree with chromatography using His tag [43]. In addition, as noticed in several studies, residual contaminants were still observable after His- [30,42] and GST-tagged [31] FN fragments purification. Conversely, here the fragment was recovered in greater than 98% purity using the double chromatography purification (Figure 2) and no other bands were visible on SDS-PAGE gel.

This observation indicates that the performed double affinity chromatography completely removes contaminants and enables the recovery of the recombinant fragment with high degree of purity.

### 2.3. Mass Spectrometry

The presence of the short peptide sequences RGD and PHSRN designed in the recombinant FN fragment (rFNIII9-10) and implicated in cell adhesion was characterized by mass spectrometry. For this purpose, the protein band (42 kDa) corresponding to CMAT-rFNIII9-10 separated by SDS-PAGE was analyzed by MALDI-TOF/TOF as described previously [44]. Because the fusion protein contains a mixture of proteins, MS/MS identification is required to identify peptides from the FN fragment. Using the Peptide Mass software (http://www.expasy.org, accessed on 22 June 2021), all masses corresponding to the theoretical cleavage of the recombinant FN fragment with trypsin, were labeled on the MS spectrum. Labeled peptides (9 peaks) are reported in the FN sequence (Figure A2). Interestingly, peptide masses corresponding to sequences including the RGD domain (1591.8 and 2470.3 m/z) and the PHSRN domain (995.5 and 2532.3 m/z) are detected on the MS spectrum. To identify these peptides, peaks of interest were submitted to MS/MS fragmentation. Three of them were unambiguously identified. MS/MS results confirm the presence of expected sequences (RGD and PHSRN) in the fusion protein (CMAT-rFNIII9-10) (Figure A2 and Table A1).

### 2.4. Affinity Binding Assay of rFNIII9/10 to α5β1 Integrins Receptors

Previous reports indicate that FN binds to integrins through the RGD site whereas a synergic PHSRN site is critical for binding to α5β1 integrins especially in [25,26]. Aiming to validate the biological activity of a produced fragment and its interaction with α5β1 integrin, affinity binding of the two proteins was investigated. Thus, to determine whether rFNIII9/10 interacts directly with integrin α5β1, we performed a binding affinity assay using the purified protein. Figure 3 shows that the binding of recombinant produced fragment to α5β1 is direct and increased in a dose-dependent way. Moreover, binding reached saturation for 0.4 µg integrins coating quantity, showing that α5β1 integrins potency for rFNIII9/10 recruitment is sensible. As we can see, the ligands induced concentration-dependent and saturable rFNIII9/10 recruitment profile. This result confirms that FNIII9/10 binds specifically and directly to the α5β1 integrins. Altogether, these results indicate that produced rFNII9/10 is an efficient ligand for integrin α5β1 cell receptors and provides significant insight into the role of the 9th and 10th type III repeats as the central cell-binding domain of FN.

As expected, produced ligand (rFNIII9/10) binds successfully to the α5β1 integrins. This result confirms that FNIII9/10 binds specifically and directly to the α5β1 integrins and provides significant insight into the role of the 9th and 10th type III repeats as the central cell-binding domain of FN.

### 2.5. rFNIII9/10 Simple Coating Promotes Cellular Adhesion

rFNIII9/10 fragment was produced with high purity reaching 95%. The presence of RGD and PHSRN sequences validated by MS/MS and the binding capacity to integrins determined by performed affinity binding assay are in excellent agreements with previous reports [17,45,46]. To further evaluate the bioactivity of this fragment, a bio-adhesive surface was designed in this study by coating a non-adhesive support with rFNIII9/10. The surface ability of promoting cellular adhesion was studied.

#### 2.5.1. Passive Adsorption of rFNIII9/10 onto Plate Surface

To evaluate the adsorbed ligand density onto polystyrene (PS) plaque wells, a simple and preferment technique based on enzyme-linked immobilization assay (ELISA) was used. Increasing concentrations of rFNIII9/10 solution were added in plate wells for 1 h and the density of adsorbed protein was determined via ELISA assay using streptavidin coupled to horseradish peroxidase (HRP). Owing to CMAT fusion partners rapid and sensitive detection of adsorbed rFNIII9/10 was enabled through the direct interaction between a streptavidin HRP and SBP tags presented in CMAT. The correlation between this method and bicinchoninic acid (BCA) quantification was confirmed as shown in Figure 4. Thus, CMAT fusion technology affords a rapid tracking method for proteins adsorption onto supports. 

At low rFNIII9/10 coating concentrations, differences in the amounts of adsorbed protein were detected. Adsorption increased with fragment concentration when less than 60 µg/mL (R^2^ = 0.75, *p*-value (H0; slope = 0) < 0.05)) and 40 µg/mL (R^2^ = 0.844, *p*-value (H0; slope = 0) < 0.05)) for 1 h and overnight treatment, respectively. Fragment surface density reached saturation at a coating concentration near 60 µg/mL for 1 h treatment (R^2^ = 0.6, *p*-value (H0; slope = 0) > 0.05)) and 40 µg/mL (R^2^ = 0.75, *p*-value (H0; slope = 0) > 0.05)) for overnight treatment. To summarize, this designed model presents a well-defined surface that allows direct functional comparison on a molar basis of rFNIII9/10 as adhesive ligand.

#### 2.5.2. FN-Mimetic Surfaces Supporting Cellular Adhesion

Based on FNIII9-10 adsorption study results, we engineered bio-adhesive surfaces presenting different densities of rFNIII9-10 in order to evaluate whether designed fragments mediate specific cellular adhesion of human gingival fibroblast (hGF) (Figure 5B).

Cellular adhesion was studied on surfaces with or without an initial rFNIII9/10 coating after 2 h of incubation in a serum-free medium. Results show an increase in a dose-dependent manner of cell adhesion to adsorbed rFNIII9/10 (Figure 5). Uncoated surfaces showed a significatively low cell attachment compared to FNIII9-10-coated surfaces where cell adhesion observed was higher (*p* < 0.01 ANOVA). As shown in Figure 5 cell adhesion to FNIII9-10 developed surfaces presented dose-dependent increases with FNIII9-10 coating concentration. The increases of adherent cell numbers probably resulted from higher sensibility of integrin cell receptor with a quantity of rFNIIII9-10. We further investigated early cell attachment at 1, 2, and 3 h on rFNIII9/10-coated well surfaces at saturating coating concentration and results showed a difference in adhesion from 1 h after seeding cells compared with uncoated wells (*p* < 0.02 ANOVA). Adhesion was significatively greater than that observed on untreated surfaces for the investigated duration (main *p* < 0.02 ANOVA) and adherent cells to rFNIII9/10 surfaces displayed well morphologies and were already well-spread after only one hour of adhesion (Figure 5A). To further investigate rFNIII9/10′s ability to promote cellular adhesion, the bioactivity of the recombinant fragment was compared with full FN. Our results show that the rFNIII9/10 exhibited equivalent levels of adhesion (Figure A3). Our findings concur with results reported by Martino et al. (2009) who obtained similar adhesion level of cells to recombinant FN cell-binding domains (FNIII9-10)-coated surfaces compared to native FN [47].

Taken together these results indicate that engineered fragment support integrin α5β1-mediated cellular adhesion. Moreover, bio-adhesive surfaces with immobilized FNIII9-10 present dose-depending levels of cellular adhesion and enable early cell attachment.

## 3. Discussion

Significant efforts are being made to further understand and control ECM-integrin interactions as it is essential for many applications including the development of biomimetic materials. In this study, bioactive fragments derived from FN-cell-binding domains was engineered in fusion with multi-tags for a simple and rapid characterization. The main findings from this study are as follows: (i) Engineered fragment encompassing the RGD motif and PHRSN synergic site enables α5β1 integrin direct interaction; (ii) support cell adhesion on coated surfaces; (iii) fusion partners used provide a time and cost-efficient simple technology for the fragment characterization as previously described. This study aimed to produce the functional sites of the FN-cell-binding domains and to develop a simple and efficient approach for its characterization and adsorption studies. FN domains have been widely expressed with fusion partners for both easy step purification using the His-tag [30] and GST-tag [26] or for enhancing bio-functionality using cadherin [31] and keratinocyte growth factor [32]. In this work, we have described for the first time the production of the cell-binding domain of human FN type III (FNIII9-10) containing the RGD motif and the synergy sequence in fusion with multi-tags (CMAT) in *E. coli*. It was initially confirmed that an engineered fragment was recovered in greater than 98% purity using the double chromatography purification using CMAT. Although the degree of purity is of particular interest for the desired application of these bioactive molecules, previous studies are less interested in this parameter. Residual contaminants were observable after His-tagged [45] and GST-tagged [26] for FN fragments purification. This was also observed in Keefe et al.’s findings who showed a lower purification degree with chromatography using His-tag [46]. In this study, the use of multi-tag fusion partners enabled a double affinity chromatography on His and SBP tag which completely removes contaminants as shown in Figure 2. This result underscores the importance of bimolecular engineering strategies and could provide a robust approach for FN fragments purification. FN central cell-binding domain has been well studied and it has been demonstrated that FNIII10-containing RGD motive binds to several integrins while FIII9 compressing PHSRN site is required for binding specifically to α5β1 integrin. Here, we confirmed this interaction by performing affinity-binding assay of α5β1 integrin using recombinant FNIII9/10 fragment as ligand. Results showed rFNIII9/10 binds α5β1 integrins successfully. Therefore, one may assume that rFNIII9/10 directly interacts with α5β1 integrins. This result is in general agreement with previous studies exploring the synergistically acting of RGD motif and PHRSN site for the α5β1 integrins-FN interaction [48,49]. The bio-adhesive activity of a recombinant fragment was then explored by tethering a non-adhesive surface with FNIII9/10 fragment to generate controlled bio adhesive interfaces. Several investigators have functionalized surfaces with FN mimetic peptides supporting cellular adhesion and demonstrated the exquisite sensitivity of α5β1 integrins binding to perturbations in the orientation of RGD-PHRSN as in the native FN [50,51,52,53]. Moreover, previous reports showed that surface chemistry modulate the amount and the conformation of adsorbed FN fragments which significantly affect its bio adhesive activity [48]. In this study, simplified ELISA was used to evaluate passively adsorbed rFNIII9/10 through SBP tag. The results showed a linear correlation of used methods with basic protein quantifications assays. A linear increase of adsorbed rFNIII9/10 with increasing coating concentration was observed for concentrations of up to 40 µg/mL and 60 µg/mL for 60 min and overnight respectively and a well-defined saturation plateau was identified for the highest concentrations suggesting that those concentrations were sufficient to saturate the plate’s surfaces (*p*-value (H0; slope = 0) > 0.05). Evaluating tethered ligand densities is an important design parameter for generating bio interfaces so that it makes the use of ELISA strategy in this study more relevant to current reports. Finally, the ability to promote cell attachment of an engineered fragment was evaluated. As a result, coating non adhesive surfaces with rFNIII9/10 was shown, as expected, to be effective in promoting cellular adhesion. Moreover, a low density of rFNIII9/10 was sufficient to increase cell adhesion compared to negative control (absence of rFNIII9/10 coating) and similar adhesion levels with native FN. The improved adhesive activities of functionalized surfaces using rFNIII9/10 fragment can be attributed to enhanced α5β1 integrin binding. It is hypothesized that the RGD motif and PHSRN are presented in the same conformation as the native FN leading to a higher receptor-binding affinity [49]. Furthermore, Petrie TA and al. (2006) showed that recombinant domains of FN (FNIII7-10) in fusion with a biotin tagging sequence at the amine terminus were still able to activate FAK phosphorylation [46]. FAK localizes to focal adhesions and activates various signaling cascades regulating cell survival, proliferation, and differentiation [54]. Petrie TA and al. obtained similar adhesion levels as reported in our study with a validated FAK activation. We can clearly see that bioadhesive properties of rFNIII9/10-engineered surfaces may be attributed to integrin–ligand bonds which have the ability to promote intracellular signaling of the recombinant fragment [55].

In summary, this study provides a general agreement with previous reports in the field of FN α5β1 integrin interactions and an alternative experimental strategy to engineer and study recombinant fibronectin domains using fusion technology. Finally, engineered rFNIII9/10 was shown to directly interact with α5β1 integrins and be efficient in promoting cell adhesion on tethering surfaces. This makes it an interesting bioactive fragment that could potentially be used to improve bio integration of tissue engineered scaffolds in biomedical devices.

## 4. Materials and Methods

### 4.1. Materials, Expression Vector and Strains

Reagents including tryptone, yeast extract, ampicillin, and agarose were purchased from Euromedex (Souffelweyersheim, France). DNA polymerase, TA Cloning kit, and gel electrophoresis were bought from Invitrogen (Waltham, MA, USA). T4 DNA ligase, oligonucleotides for gene amplification, isopropyl β-*D*-1-thiogalactopyranoside (IPTG), δ-aminolevulinic acid (ALA) and molecular weight markers were purchased from Sigma (St. Louis, MO, USA). Restriction endonucleases and enzymatic quantification reagent bicinchoninic acid (BCA) were from Thermo Scientific (Waltham, MA, USA). NucleoSpin Plasmid was obtained from Macherey-Nagel (Düren, Nordrhein-Westfalen, Germany). Streptactin column was purchased from Novagen and NI-NTA agarose from Sigma (St. Louis, MO, USA). Expression vector pET15b-CMAT used to produce the recombinant rFN was previously developed in our laboratory [40]. *E. coli* strains Top10F’ (Invitrogen, Waltham, MA, USA) and BL21 (DE3) pRARE (Novagen, Darmstadt, Germany) were used for all subclonings and expression studies, respectively. All enzymatic quantification reagents including Streptavidin-Peroxidase (Ultrasensitive), 3,30,505-tetramethylbenzidine (TMB) liquid substrate system, stop reagent for TMB substrate and Tween 20 were also purchased from Sigma (Steinheim, Germany). HisGrab TM Nickel-coated 96-well plates were obtained from pierce (Rockford, IL, USA). StrepTrap TM HP and HisTrap TM HP columns used for the purification of recombinant proteins were procured from GE Healthcare (Uppsala, Sweden). NZY Auto-Induction LB medium (powder) was bought from NZYTech gene and enzyme (Lisboa, Portugal). Other reagents including Bacto tryptone, ampicillin, agarose, and yeast extract were bought from Euromedex (Souffelweyersheim, France).

### 4.2. Expression Vector pET15b-CMAT-rFN

DNA sequence encoding FNIII9-10 was standard PCR amplified using primer 1 (5′- cacgtgtaggtcttgattccccaactgg) and primer 2 (5′-gctcagcttaattgttcggtaattaatggaaat). The forward and the reverse primers were designed to add a 5′ PmlI and a 3′ BlpI restriction sites, respectively. PCR cycles were as follows: denaturation at 94 °C for 1 min, annealing at 58 °C for 1 min, and extension at 72 °C for 2 min for 30 cycles. The PCR amplicon was first subcloned into pCR2.1 for an optimized cloning then cloned into pET15b in phasis with the previously inserted CMAT sequence. This fusion partner contains a colored tracking marker (cytochrome b5) and two affinity purification tags (10His and SBP). To allow removal of the CMAT partner, a Tobacco Etch Virus (TEV) cleavage site was inserted to recover the rFN from the fusion protein. The expression vector containing the recombinant CMAT-rFN protein was sequenced to validate the cloning step (Sanger method, GATC-biotech, Mulhouse, France).

### 4.3. Protein Expression

*E. coli* BL21 (DE3) pLysS strain transformed with the recombinant plasmid was cultivated overnight on LB agar plate containing 100 µg/mL ampicillin and 34 µg/mL chloramphenicol at 37 °C. A starter culture (10 mL) prepared from one isolated colony was used to inoculate 100 mL of fresh LB medium supplemented with ampicillin and chloramphenicol. OD600 nm was standardized to 0.1 and shaking flask was incubated at 37 °C, 225 rpm until biomass reached the early log phase growth. At OD600 nm around 0.5, induction was performed at 25 °C with 0.5 mM IPTG in the presence of 2 mM ALA. Samples at different time points were analyzed for OD600 nm and OD400 nm (see qualitative and quantitative monitoring section). After overnight incubation, cells were harvested by centrifugation at 4000 rpm for 15 min (4 °C) then washed with PBS. The cell pellet was re-suspended in 10 mL of phosphate buffered saline (PBS), pH 7.4 and 0.5% Triton X-100 supplemented with 1mg/mL of Lysozyme and a protease inhibitor cocktail (Roche Diagnostics, Basel, Switzerland) and incubated 1 h at 4 °C, then supernatant and lysate pellet fractions were separated by centrifugation at 13,000 rpm for 30 min.

### 4.4. First Purification Step Based on 10His Tag

About 10 mL of the soluble 10His-tagged protein fraction extracted from induced culture was mixed with 250 µL equilibrated His-select^®^ Nickel Affinity gel for 1 h at 4 °C. The charged resin was packed in a 5 mL column and washed with three column volumes of PBS (pH 7.4) containing 15 mM Imidazole to remove contaminating *E. coli* proteins. The elution was performed using 250 µL of PBS (pH 7.4), 250 mM imidazole.

### 4.5. Second Purification Step Based on SBP Tag

The eluted fractions from the first purification step were pooled, diluted five times in PBS, and applied onto Strep.Tactin^®^ SpinPrep™columns previously equilibrated with 500 µL wash buffer (150 mM NaCl, 100 mM Tris-HCl, 1 mM EDTA, pH 8.0) according to the manufacturer’s instructions. Non-specifically bound proteins were removed by four washing steps, each with 100 µL of wash buffer and the SBP-tagged protein was eluted with 150 µL of elution buffer (150 mM NaCl, 100 mM Tris–HCl, 1 mM EDTA, 2 mM D-biotin, pH 8.0). The final purified protein was dialyzed and frozen for storage at −80 °C until further analysis.

### 4.6. Qualitative and Quantitative Monitoring

Real-time monitoring of proteins production and purification was qualitatively assessed by following the appearance of a red coloration during the handling processes. The total protein amounts at different post-induction times were determined using a BCA assay kit. For the monitoring of the rFNIII9/10 tagged protein, the absorbance of the lysate fractions was measured at 400 nm [40]. Results from OD400 nm quantification were compared with BCA quantification and a relationship between the quantity of fusion proteins (rFNIII9/10) and the total soluble proteins was established.

### 4.7. SDS-PAGE Analysis and Quantification

Fusion protein expression, solubility, and purification steps were performed by SDS-PAGE on 4–12% Bis-Tris gel as previously described [40]. Gels were captured with ImageQuant 350 camera and bands were quantified with ImageQuant TL software (GE Healthcare, Chicago, IL, USA). Recombinant fusion proteins were identified by ELISA using the streptavidin–peroxidase conjugated to HRP and the TMB substrate and confirmed by mass spectrometry.

### 4.8. Mass Spectrometry Analysis

The band to be identified was excised from gel and digested in-gel with trypsin using the automated Proteineer SPII (Bruker Daltonik, Billerica, MA, USA) and Ettan Digester (GE Healthcare, Chicago, IL, USA) robots, following the manufacturer’s instructions. Digests were mixed with saturated solution of cyano-4-hydroxycinnamic acid (CHCA) matrix in 50% acetonitrile/2% formic acid and spotted on a MALDI sample plate. The MALDI-TOF/TOF (UltrafleXtreme, Bruker Daltonics, Billerica, MA, USA) instrument was operated in the positive ion mode and controlled by the Compass for Flex software, version 1.4 (FlexControl 3.4, FlexAnalysis 3.4, Bruker Daltonics); 4000 laser shots were accumulated per spectrum in the MS and MS/MS modes. The MS spectra were externally calibrated using the peptide calibration standard mixture (Bruker Daltonics, Billerica, MA, USA). Peptides of interest were selected for MS/MS according to MS data. The data obtained by MS and MS/MS were submitted to the SwissProt/UniProtKB database (http://www.expasy.org, accessed on 22 June 2021) using the MASCOT software (Matrix Science, Torrance, CA, USA). The search parameters used were as follows: Homo sapiens taxonomy; fixed carbamidomethylation of cysteine residues; variable oxidation of methionine residues; peptide mass tolerance of 30 ppm and fragment-ion mass tolerance of 0.5 Da. maximum was allowed; one enzymatic missed cleavage maximum was allowed; trypsin as digestion compound. Mascot MS/MS ion score >30 indicate identity (*p* < 0.05).

### 4.9. Affinity Binding Test of rFNIII9/10 to α5β1 Integrins Receptors

To investigate the interaction of rFNIII9/10 with α5β1 integrins, a binding affinity assay was performed in triplicate. Wells of a 96-well plate were coated overnight at 4 °C with 100 µL of α5β1 integrins diluted in PBS at several concentrations. Wells were then washed with 0.02% tween PBS three times and blocked with 1% of denatured BSA for 2 h at 37 °C. About 3.2 µg of rFNIII9/10 prepared in binding buffer (50 mM trisHCL (pH 7.4), 100 mM NaCL, 2 mM CaCl2, 1 mM MgCl2, 1 mM MnCl2) was added and allowed to bind for 2 h at 37 °C. After 2 h of incubation, the rFNIII9/10 solution excess was removed, and plates were washed three times with 200 µL of PBS 0.05% Tween-20. Streptavidin-Peroxidase at 1:5000 dilution (in PBS, 0.05% Tween-20) was then added and incubated for 1 h at room temperature under gentle agitation. After removing the streptavidin-peroxidase solution, plates were washed again as described above. For enzymatic activity measurement, 150 µL of TMB substrate was added and plates were incubated for 30 min. After acidification with 150 µL of 0.5 M H2SO4 stopping solution, the absorbance of each well was read with a microplate reader at 450 nm.

### 4.10. Quantification of Passively Adsorbed rFNIII9-10 Using Enzyme-Linked Immobilization Assay

rFNIII9/10 was passively adsorbed onto polystyrene wells surface to determine the saturating levels and to study its adsorption proprieties. PS wells were rinsed twice with PBS and then incubated with serial dilutions (1:2) of protein, starting at a concentration of 10 mg/mL for 1 h or overnight at 37 °C. Unbound proteins were removed by washing several times with PBS. Adsorbed rFNIII9/10 was quantified by ELISA as described previously [30]. Briefly, wells were washed three times with 200 µL of PBS +0.05% Tween-20. Streptavidin-peroxidase at 1:5000 dilution (in PBS +0.05% Tween-20) was then added and incubated for 1 h at room temperature under gentle agitation. After removing the streptavidin-peroxidase solution, plates were washed again as described above. For enzymatic activity measurement, 150 µL of TMB substrate was added and plates were incubated for 30 min. After acidification with 150 µL of 0.5 M H_2_SO_4_ stopping solution, the absorbance of each well was read with a microplate reader at 450 nm. For BCA assays, adsorbed proteins were solubilized from the surface using 50 µL 4 M Guanidine-HCL for 1 h. Total of 200 µL of BCA working reagent was added and incubated for 30 min at 37 °C. Absorbance was measured at 562 nm.

### 4.11. Cell Adhesion Assay

Human gingival fibroblasts (hGF) were used in this study. hGF are primary human cells that adhere and spread on human FN-coated surfaces using α5β1 integrins [39,56,57]. Cells were cultivated in DMEM supplemented with 10% fetal bovine serum and 1% penicillin/streptomycin at 37 °C in 5% CO_2_. Non-tissue culture plates (48-wells) were coated with recombinant fibronectin proteins diluted in PBS or full FN (Fibronectin Human, Plasma, Thermo Fisher Scientific) at the indicated concentrations for 60 min at 37 °C and at saturating coating concentration for overnight incubation. Unbound protein was removed, and wells were washed with PBS and blocked with 1% denatured BSA for 1 h. HGF cells were seeded at 1.4 × 10^5^ cells/cm^2^ and allowed to adhere for 2 h at 37 °C, 5% CO_2_ in serum-free medium. After incubation, cells were washed with DPBS and labeled with calcein-AM, a cell-membrane permeable fluorescent marker (485 nm/535 nm), at 4 mm in DPBS+2 mm dextrose for 45 min, to calculate the density of adherent cells. In addition, cells were photographed with a Nikon TE-300 fluorescence microscope at 4.5 magnification.

### 4.12. Statistics

Each experiment was run in triplicate. Results were analyzed using the ANOVA method to determine the variance and the *p* value. Data are reported as mean ± standard error. For the plateau of the saturation curve evaluation, the (right-tailed) F probability distribution (degree of diversity) for two data sets (F.DIST.RT function) was used to study hypothesis; H0; slope = 0.

## 5. Conclusions

This study investigates the development of innovative bio-adhesive ligand derived from FN-cell-binding domains including RGD sequence and PHSRN synergic motif in fusion with multi-tags providing a rapid and efficient simple technology for protein characterization. Designed bio-adhesive motif enables direct interaction with α5β1 integrins. Moreover, surfaces were passively coated with FNIII9-10-supported cell adhesion in a dose-dependent level. These results provide an innovative approach for performing bio-adhesive ligand with simple and rapid characterization techniques. In fact, results provide a biological active fragment displaying a comparable adhesive activity to full-length fibronectin which make it an interesting candidate for surface tethering. In addition, this strategy incorporates an interesting tag (SBP) offering a high affinity to streptavidin paving the way for the development of an oriented immobilization technology onto streptavidin layer coated onto supports for further controlled bioadhesive interfaces studies.

## Figures and Tables

**Figure 1 ijms-22-07362-f001:**
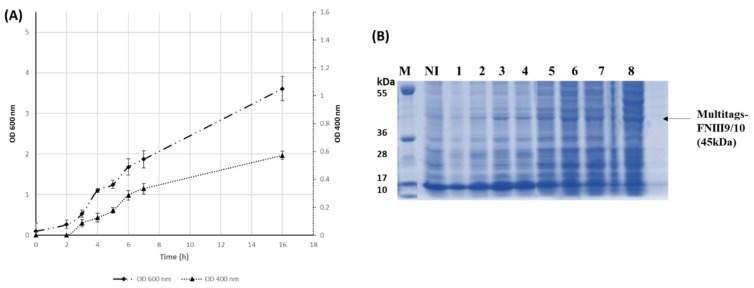
Recombinant FNIII9/10 production: (**A**) real time monitoring of Multitags-FNIIII9/10 production: Protein was expressed in *E. Coli BL21 pLysS* induced with 0.5 IPTG at 25 °C (**A**) represents optical density (OD) profile at 600 nm (•) for biomass evolution and at 400 nm (▲) for recombinant protein soluble expression, (**B**) represents SDS–PAGE analysis of the time course protein expressions. Proteins were separated on 12% Bis-Tris gels (Invitrogen, Waltham, MA, USA). M: protein molecular weight marker. Ni: soluble fraction from cells withdrawn at the time of induction. Lanes 1–8: soluble fractions from cells withdrawn at 1, 2, 3, 4, 5, 6, 7, and 16 h after induction, respectively.

**Figure 2 ijms-22-07362-f002:**
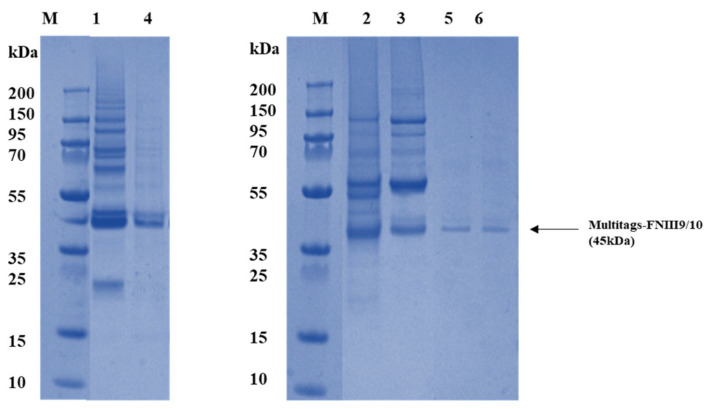
Purification analysis of rFNIII9/10; SDS_PAGE analysis of purified rFNIII9/10 fragment. Proteins were separated on 12% Bis-Tris gels (Invitrogen, Waltham, MA, USA): M: protein molecular weight markers. Lanes 1, 2, and 3: purified proteins using His tag, Lane 4–6: purified proteins with the double purification using His tag and SBP tag as described in the “methods” section.

**Figure 3 ijms-22-07362-f003:**
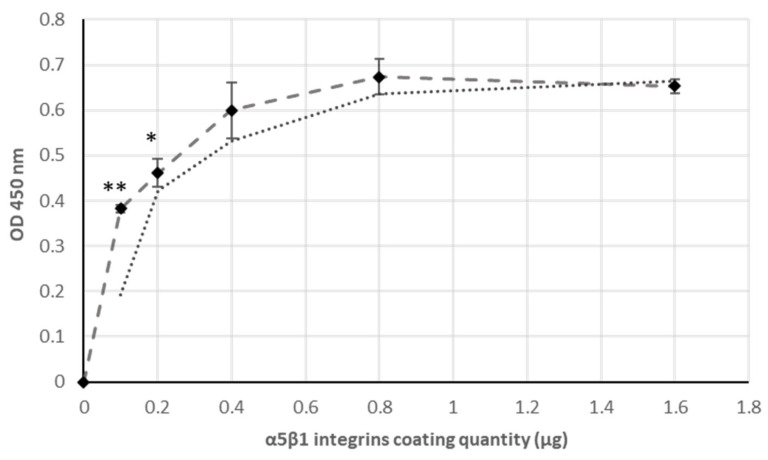
rFNIII9:10—α5β1 integrin affinity binding assay: Integrin α5β1 specifically binds to rFNIII9/10: Binding profile of coated α5β1 integrins to rFNIII9/10; figure represent α5β1 integrins coating quantity as a function of OD 450 nm: rFNIII9/10 (3, 2 µg) was incubated with increasing coating concentrations of recombinant α5β1 integrins up to 16 µ/mL. Binding complex was evaluated owing to SBP tag bound to Streptavidin HRP, as described in the experimental procedure and graphical abstract. Experiments were performed in triplicate, and each assay point in the binding curves represents mean ± SD of three measurements (Main *p* ANOVA < 0.05). * Significantly different *p* < 0.05 (ANOVA) and ** high significantly different *p* < 0.02 (ANOVA).

**Figure 4 ijms-22-07362-f004:**
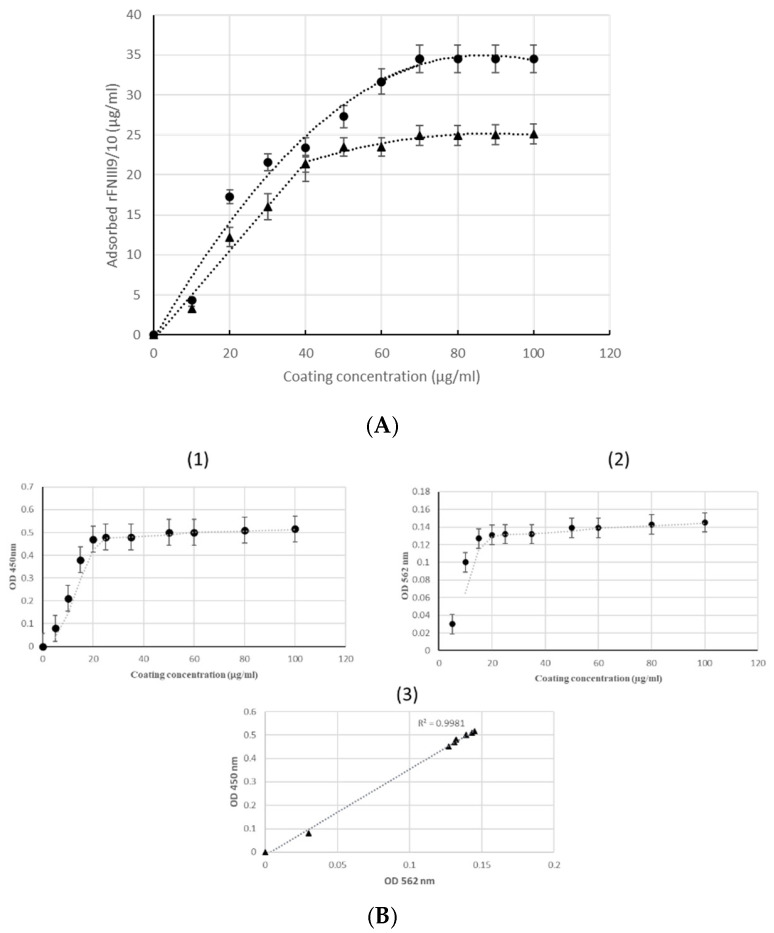
Adsorption study of FNIII9/10 onto well plate surface: (**A**) Passively adsorbed proteins as a function of coating concentration for 60 min (▲) and overnight (•) incubation; (**B**) correlation between BCA and enzyme-linked immobilization assay-based method used for adsorbed protein quantification R^2^ = 0.9948: Briefly, (**1**) adsorbed proteins were quantified by enzyme-linked immobilization assay [40,41] based method owing to SBP tag, and (**2**) BCA kit was used to quantify adsorbed proteins; (**3**) correlation between enzyme-linked immobilization assay and BCA; OD450 nm as a function of OD562 nm. (Main *p* ANOVA < 0.05).

**Figure 5 ijms-22-07362-f005:**
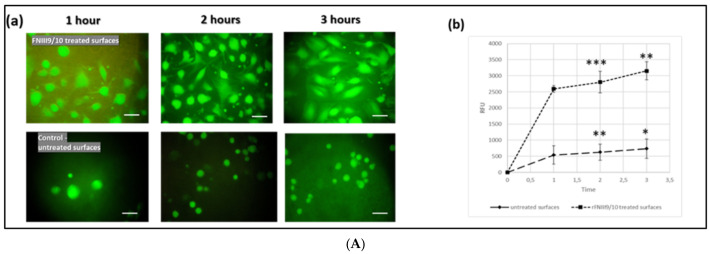
Cellular adhesion study on rFNIII9/10-coated surfaces: hGF cell adhesion to rFNIII9/10-coated surfaces showing an early cell attachment and a comparable dose-dependent level. (**A**) Early cell adhesion for almost 3 h of incubation. Early cell adhesion was evaluated in serum-free medium onto uncoated and rFNIII9/10-coated surfaces (60 µg/mL coating concentration) for 1-, 2-, and 3-h incubation, cells were labeled with Calcein AM fluorescence marker. (**a**) Micrographs of fluorescence-labeled adherent cells at different incubation times studied (1, 2, and 3 h) for negative control and rFNIII9/10-treated surfaces (magnification scale 40×). (**b**) Adherent cell assessment: relative fluorescence units (RFU) as a function of time. (**B**) rFNIII9/10 surface density effect on hGF adhesion; adhesion was studied in serum-free medium onto different rFNIII9/10-coated surfaces: [0 µg, 5 µg, 10 µg, and 20 µg adsorbed rFNIII9/10] for 2 h adhesion; cells were labeled with calcein AM fluorescence marker; (**a**) adherent cell assessment; (**b**) image of adherent cells without labeling for uncoated and rFNIII9/10-coated surfaces (magnification scale 10X). (**c**) Image of calcein-labeled hGF cells adhering to treated/not treated surfaces for 2 h (magnification scale 20×). Data are expressed as mean relative fluorescence units (RFU) of triplicate wells ± SEM. * Significantly different *p* < 0.05 (ANOVA), ** high significantly different *p* < 0.02 (ANOVA), *** very high significantly different *p* < 0.01 (ANOVA).

## Data Availability

The original data and the statistical analyses and the data that were cited as “not shown” can be obtained from the corresponding author upon request.

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
