# Peer review of "Engineering of Bio-Adhesive Ligand Containing Recombinant RGD and PHSRN Fibronectin Cell-Binding Domains in Fusion with a Colored Multi Affinity Tag: Simple Approach for Fragment Study from Expression to Adsorption"

_ijms, 2021, doi:10.3390/ijms22147362_

Round 1

Reviewer 1 Report

In the manuscript entitled “Engineering of bio-adhesive ligand containing recombinant RGD and PHSRN fibronectin cell binding domains in fusion with a colored multi affinity tag : simple approach for fragment study from expression to adsorption”, the authors use the recombinant protein technology to produce the cell binding domains of the ECM protein fibronectin (FN) that is known to bind integrins and in particular integrin alpha5beta1. The recombinant fragment constituted by the 9th and 10th fibronectin type III repeat domains (FNIII9-10), containing the Arg-Gly-Asp (RGD) and Pro-His-Ser-Arg-Asn (PHSRN) binding sites, was expressed in fusion with a Colored Multi Affinity Tag (CMAT) that was introduced in order to simplify the purification and characterization steps.

The recombinant fusion protein CMAT-FNIII9/10 demonstrated to improve the real time monitoring of its production and the introduction of two affinity purification tags: 10His (polyhistidine) and SBP (Streptavidine Binding Protein) allowed to obtain it in an high purity degree. The ability of the FNIII9-10 fragment to bind integrin alpha5beta1 was demonstrated, moreover plate coating with rFNIII9-10 proved to mediate cellular adhesion of human Gingival Fibroblast (hGF).

Engineered ECM proteins can be useful both in further understand and control ECM-integrin interactions and in their cost effective production. The approach used by the authors that designed a “simplified” FN type III repeat (FNIII9-10) and introduced a multi affinity tag to improve its purification, fulfill both these requests.

Nevertheless, some aspects should be improved before publication.

First of all, the manuscript would take advantage by a comparison between the performance of recombinant FNIII9-10 and that of human FN both in alpha5beta1 direct binding evaluation and in cell adhesion assays.

Moreover, the author state that “The improved adhesive activities of functionalized surfaces using rFNIII9/10 fragment can be attributed to enhanced α5β1 integrin binding. It is hypothesized that the RGD motif and PHSRN are presented in the same conformation as the native FN leading a higher receptor binding affinity [49].” (pages 9-10, line 333-337); nevertheless experiments aimed to prove this statement are not reported. A comparison with a cell line not expressing integrin alpha5beta1, or a blocking step with an alpha5beta1 binding molecule or an antibody could be helpful. On the contrary, cell adhesion to rFNIII9/10 functionalized surfaces could also be ascribed to other integrins or even to different factors. Data about alpha5beta1 integrin expression by human Gingival Fibroblast (hGF) should also be added.

Other points to be addressed are:

-Page2, line 62: the authors probably mean integrin αvβ3. Integrin α5β3 does not exist.

-Page3, Figure 1: resolution of the figure must be improved

-Page4, Figure 2: reference column M seams to belong to a different gel/experiment and seams the same for the two images.

-Page6, Figure 4(I): the fitting of the upper curve seams to stop and re-start at 60 µg/ml conc.

-Page 9: the authors state “FN central cell binding domain has been well studied and it has been demonstrated that FNIII10 containing RGD motif binds to several integrins while FIII9 compressing PHSRN site is required for binding specifically α5β1 integrin” (lines 306-308) and “Several investigators have functionalized surfaces with FN mimetic peptides supporting cellular adhesion and demonstrate the exquisite sensitivity of α5β1 integrins binding to perturbations in the orientation of RGD-PHRSN as in the native FN.”(lines 316-318). Bibliographic references should be added.

-References should be formatted according to the journal style, moreover ref. 42 is not complete.

The manuscript need to be revised for typos and grammar correction.

Author Response

Reviewer 1

Object: revised article

Dear reviewer 1 ,

We would like to thank Reviewer 1 for the comments and suggestions. Further justifications and explanations of the results have helped us to improve the manuscript; we have replied to each comment as follows:

First of all, the manuscript would take advantage by a comparison between the performance of recombinant FNIII9-10 and that of human FN both in alpha5beta1 direct binding evaluation and in cell adhesion assays.

  • We have already evaluated the ability of produced fragment to imitate native FN by comparing their potential to promote cellular adhesion onto coated plates with native FN. We added these results to our manuscript for submission. Please consult page 17 and page 9, line 285 to page 10, line 290.

Moreover, the author state that “The improved adhesive activities of functionalized surfaces using rFNIII9/10 fragment can be attributed to enhanced α5β1 integrin binding. It is hypothesized that the RGD motif and PHSRN are presented in the same conformation as the native FN leading a higher receptor binding affinity [49].” (pages 9-10, line 333-337); nevertheless experiments aimed to prove this statement are not reported. A comparison with a cell line not expressing integrin alpha5beta1, or a blocking step with an alpha5beta1 binding molecule or an antibody could be helpful. On the contrary, cell adhesion to rFNIII9/10 functionalized surfaces could also be ascribed to other integrins or even to different factors. Data about alpha5beta1 integrin expression by human Gingival Fibroblast (hGF) should also be added.

  • Data about alpha 5 beta1 integrin expression by hGF were added* (page 17, line 499).

*Kim, E.-C.; Lee, D.Y.; Lee, M.-H.; Lee, H.J.; Kim, K.-H.; Leesungbok, R.; Ahn, S.-J.; Park, S.-J.; Yoon, J.-H.; Jee, Y.-J.; et al. The Effect of Fibronectin-Immobilized Microgrooved Titanium Substrata on Cell Proliferation and Expression of Genes and Proteins in Human Gingival Fibroblasts. Tissue Eng. Regen. Med. 2018, 15, 615–627, doi:10.1007/s13770-018-0153-7.

*Oates, T.W.; Maller, S.C.; West, J.; Steffensen, B. Human Gingival Fibroblast Integrin Subunit Expression on Titanium Implant Surfaces. J. Periodontol. 2005, 76, 1743–1750, doi:10.1902/jop.2005.76.10.1743.

Other points to be addressed are:

-Page2, line 62: the authors probably mean integrin αvβ3. Integrin α5β3 does not exist.

  • The mistake was corrected (page 2, line 62) .

-Page3, Figure 1: resolution of the figure must be improved

  • Figure 1 has been improved (page 3) .

Page4, Figure 2: reference column M seams to belong to a different gel/experiment and seams the same for the two images.

  • Data about the gels were uploaded with manuscript. It belongs to the same gel/ experiments for the column M and we have only reorganized the position of analyzed fractions. Figure 2 has been improved (page 5) and we added here the document with gels without modification already uploaded for submission. Please consults document of original gels: figure 2 original non adjusted photos.

-Page6, Figure 4(I): the fitting of the upper curve seams to stop and re-start at 60 µg/ml conc.

  • We did this in order to show the saturation level as performed in chemical adsorption studies. The curve has been modified without stop point (page 7).

-Page 9: the authors state “FN central cell binding domain has been well studied and it has been demonstrated that FNIII10 containing RGD motif binds to several integrins while FIII9 compressing PHSRN site is required for binding specifically α5β1 integrin” (lines 306-308) and “Several investigators have functionalized surfaces with FN mimetic peptides supporting cellular adhesion and demonstrate the exquisite sensitivity of α5β1 integrins binding to perturbations in the orientation of RGD-PHRSN as in the native FN.”(lines 316-318). Bibliographic references should be added.

  • The following references* have been added (page 10 line 327, line 332) :

*Feng, Y.; Mrksich, M. The Synergy Peptide PHSRN and the Adhesion Peptide RGD Mediate Cell Adhesion through a Common Mechanism. Biochemistry 2004, 43, 15811–15821, doi:10.1021/bi049174+.

*García, A.J.; Schwarzbauer, J.E.; Boettiger, D. Distinct Activation States of Alpha5beta1 Integrin Show Differential Binding to RGD and Synergy Domains of Fibronectin. Biochemistry 2002, 41, 9063–9069, doi:10.1021/bi025752f.

*Grant, R. P., Spitzfaden, C., Altroff, H., Campbell, I. D. & Mardon, H. J. Structural requirements for biological activity of the ninth and tenth FIII domains of human fibronectin. J. Biol. Chem. 272, 6159–6166 (1997).

*Garc!ıa AJ, Schwarzbauer JE, Boettiger D. Distinct activation states of alpha5beta1 integrin show differential binding to RGD and synergy domains of fibronectin. Biochemistry 2002;41: 9063–9. [

*Adsorption of Fibronectin Fragment on Surfaces Using Fully Atomistic Molecular Dynamics Simulations Evangelos Liamas 1,2 , Karina Kubiak-Ossowska 2 , Richard A. Black 3 , Owen R.T. Thomas 1 , Zhenyu J. Zhang 1,* and Paul A. Mulheran 2,*

*Zollinger, A. J., & Smith, M. L. (2017). Fibronectin, the extracellular glue. Matrix Biology, 60, 27-37.

-References should be formatted according to the journal style, moreover ref. 42 is not complete.

  • References have been revised by using ‘id: multidisciplinary digital publishing institute citation’ style from Zotero application.

The manuscript needs to be revised for typos and grammar correction.

  • Grammar and syntax have been revised by English language professor. Our manuscript was submitted to our colleague Mrs Rozalia Czak, English lecturer at EBI, for language proof reading. Please find enclosed the certificate in the supplementary file.

Reviewer 2 Report

This report builds on an existing technique developed by the laboratory. There is not anything new or novel in its approch to the use of the technology or the peptide fragments produced. While there is a considerable amount of work here and worthwhile showing more versatility in the technique I do not believe in this format or journal it is appropriate. 

There are numerous errors in the manuscript e.g. 

Low-quality images (e.g. unreadable figure 1)

missing reference line 169

Motives should be motives line 36

Author Response

Reviewer 2

Object: revised article

Dear reviewer 2,

We would like to reply to each comment of reviewer 2 as follows:

This report builds on an existing technique developed by the laboratory. There is not anything new or novel in its approch to the use of the technology or the peptide fragments produced. While there is a considerable amount of work here and worthwhile showing more versatility in the technique I do not believe in this format or journal it is appropriate.

  • We would like to thank reviver 2 for the comments and suggestions. Although we had previously reported on CMAT technology, the aspect investigated in this study is novel. In our previous reports we evaluated the ability of CMAT for real time monitoring of recombinant protein expression and for high level purification. Here, we described for the first time its potential to evaluate protein adsorption and detection on coated surfaces. The description of fusion partner has already been reported but this novel application has not. Protein detection and adsorption evaluation are among the most widely investigated features in protein science and basic used techniques are expensive and time consumer. The described approach based on SBP and streptavidin interaction may offer an efficient and attractive solution for similar studies.

There are numerous errors in the manuscript e.g.

  • Grammar and syntax have been revised by English language professor. Our manuscript was submitted to our colleague Mrs Rozalia Czak, English lecturer at EBI, for language proof reading. Please find enclosed the certificate in the supplementary file.

Low-quality images (e.g. unreadable figure 1)

  • Figure 1 has been improved (page 3)

missing reference line 169

  • Reference has been added

Motives should be motives line 36

  • Motives have been added

Reviewer 3 Report

The manuscript by Abla et al. describes the design and production of functional adhesive ligand containing recombinant FN 9th and 107 10th type III repeats (rFNIII9/10) including RGD sequence and PHSRN synergic site using gene cloning and plasmid construction techniques. The paper was very interesting and demonstrated the possible application of using the recombinant sequence for cell adhesion. However there are several issues which should be taken into consideration prior to publication:

  • Figure 4 needs reorganizing, the layout and legend are a bit confusing.
  • For the adhesion assay, I will advise to use cell repellent culture plates and not Non-tissue culture plates since it is evident the fibroblast cell display some attachment to these culture plates.
  • I suggest to add additional control group, such as pure fibronectin, to the cell adhesion assay.
  • In the manuscript the authors claim there is a synergistic effect between RGD and PHSRN. It will be interesting to include an additional experiment demonstrating this effect.
  • It will be nice to add surface characterization (SEM or AFM) of the coated plate.

Author Response

Reviewer 3

Object: revised article

Dear reviewer 3,

We would like to thank Reviewer 3 for the valuable comments and the deep analysis of our manuscript. It has helped us to improve the manuscript; we have replied to each comment as follows:

  • Figure 4 needs reorganizing, the layout and legend are a bit confusing.
  • Figure 4 has been reorganized. Please consult pages 8-9
  • For the adhesion assay, I will advise to use cell repellent culture plates and not Non-tissue culture plates since it is evident the fibroblast cell display some attachment to these culture plates.
  • We totally understand the editor’s recommendation and would like to thank you for the suggestion. Control without FNIII9/10 coating showed here a very low hGF adhesion. We will include your recommendation into our future work.
  • I suggest to add additional control group, such as pure fibronectin, to the cell adhesion assay.
  • Control group with native human FN has been added. In fact, we already had evaluated the ability of produced fragment to imitate native FN by comparing its potential to promote cellular adhesion onto coated plate with native FN. We added those results to our manuscript for submission (Appendix A3). Please consult page 17 and page 9, line 285 to page 10, line 290.
  • In the manuscript the authors claim there is a synergistic effect between RGD and PHSRN. It will be interesting to include an additional experiment demonstrating this effect.
  • We would like to thank the reviewer 3 for the recommendation. We further improved the manuscript by adding argumentative reports to support our claim. Please consult page 10 from Line 327 to Line 332.

*Feng, Y.; Mrksich, M. The Synergy Peptide PHSRN and the Adhesion Peptide RGD Mediate Cell Adhesion through a Common Mechanism. Biochemistry 2004, 43, 15811–15821, doi:10.1021/bi049174+.

*García, A.J.; Schwarzbauer, J.E.; Boettiger, D. Distinct Activation States of Alpha5beta1 Integrin Show Differential Binding to RGD and Synergy Domains of Fibronectin. Biochemistry 2002, 41, 9063–9069, doi:10.1021/bi025752f.

  • It will be nice to add surface characterization (SEM or AFM) of the coated plate.
  • Following the reviewer 3 recommendation, we are looking for partners equipped with these materials to do these experiences. We will include your recommendation into our future work and our forthcoming publications

Round 2

Reviewer 1 Report

All the comments and suggestions raised by this referee have been addressed in the revised version. The present manuscript is now suitable to be published in International Journal of Molecular Sciences.

Minor: Page 10. Line289: the authors cite “Martino et al. (2009)” but reference 56 refers to “Wozniak et al. (2004)”.